# Dynamical Edge Modes in Yang-Mills Theory

**Adam Ball, Luca Ciambelli**

*Perimeter Institute for Theoretical Physics,*
*31 Caroline St. N., Waterloo ON, Canada, N2L 2Y5*

*E-mail:* aball1@perimeterinstitute.ca, ciambelli.luca@gmail.com

ABSTRACT: We study the symplectic structure and dynamics of Yang-Mills theory in the presence of a boundary. We introduce a decomposition of the fields on a Cauchy slice such that the symplectic form splits cleanly into independent bulk and edge parts. However, we find that the dynamics inherently couples these two symplectic sectors, a feature arising from the non-abelian nature of the gauge group. This is shown by extending to Yang-Mills theory the dynamical edge mode boundary condition recently introduced in Maxwell theory. We finish with analyses of the weak-field expansion and the horizon limit, finding in the latter case that the dynamical interplay between bulk and edge degrees of freedom persists.

# 1 Introduction

As intuition suggests, the properties of a physical system on a constant-time slice can be analyzed by partitioning the slice into smaller regions and studying how the regions interact at their boundaries through cutting and gluing procedures. In gauge theories, the inherently nonlocal nature of the gauge constraints requires a careful examination of the dynamical fields at the boundaries to ensure gauge covariance. These boundary fields, often referred to as edge modes, have garnered increasing attention within the scientific community. Notably, edge modes have been identified as essential in gauge theories for correctly accounting for entanglement entropy across boundaries. This has been appreciated in abelian gauge theory [1–9], in lattice gauge theory [10–12], and to a lesser extent in non-abelian gauge theory [13–22].

Edge modes have proved instrumental in understanding the gravitational degrees of freedom as well. Gravitational edge modes have been studied in [15, 23–35]. In the framework of the corner proposal in [15, 36–40] (see also the reviews [41–43] and references therein), edge modes are fundamental data sourcing the Noether charges at corners.[1] Therefore edge modes are naturally related to (quantum) reference frames [44–46]. Furthermore, similar to gauge theories, having localized degrees of freedom at the edge is necessary to address the problem of bulk factorization [47–51]. The importance of edge modes has also been discussed both in AdS and in flat holography [52–57].

The common thread among these diverse applications of edge modes is the necessity of incorporating localized fields at the boundary when analyzing local subsystems in gauge theories and gravity. These boundary fields are indispensable for restoring gauge covariance, ensuring the correct description of the subregion's Hilbert space, and accurately accounting for entanglement entropy. Given their importance, the corner proposal postulates that they are fundamental ingredients of any gravity-quantization scheme. In [7], the authors initiated a far-reaching analysis of suitable boundary conditions and decomposition of fields in the presence of edges. They successfully applied a boundary condition, called the *dynamical edge mode* (DEM) boundary condition, to Maxwell theory [7] and to $p$-form gauge theories [8]. The final aim of these works has been to carefully separate the bulk physics from the edge physics, both at the symplectic and dynamical levels.

In this work, we upgrade the analysis proposed in [7] to non-abelian gauge theories. We find a suitable field decomposition leading to a bulk-edge split of data,[2] and compute the Poisson brackets among all elementary fields. In spite of this split, we observe that the dynamics mixes bulk and edge physics, due to the non-abelianity of the gauge group. Our results provide important clues as to whether and how a similar split plays out in full non-perturbative gravity, which is the ultimate goal of this line of research. Yang-Mills and gravity both have non-abelian symmetry groups, so the successful split of phase space herein makes us optimistic for gravity. Furthermore, despite the failure of the Yang-Mills Hamiltonian to split, the gravitational Hamiltonian is famously a pure boundary term, so its split seems automatic.

---

[1]In this framework edges are typically referred to as corners.
[2]Here by "bulk" we mean the interior of the Cauchy surface.

The paper is organized as follows. We start in section 2 reviewing the basic ingredients of Yang-Mills theory and the conventions employed. We then propose in section 3 the field decomposition leading to the symplectic bulk-edge split, and compute the Poisson brackets. We study the dynamics for a timelike boundary in section 4, showing how it inevitably mixes the bulk and edge data. Selected topics are then presented in section 5: the fate of the dynamical mixing in the weak field expansion, and the horizon limit, in which we still find a non-trivial dynamical mixing. We relegate to appendix A some calculations relevant for this last point. We eventually gather concluding remarks in section 6.

## 2   Review of Yang-Mills

We work with classical Yang-Mills theory on a $D$-dimensional Lorentzian manifold $M$. We initially take $M$ to be a causal diamond, i.e. the causal domain $\mathcal{D}(\Sigma)$ of some spatial hypersurface $\Sigma$, although later in section 4.1 we will study the case when $M$'s boundary is timelike. Our gauge group is $G$, and we write $\mathfrak{g}$ for the corresponding Lie algebra. The gauge field $A_\mu$ is a $\mathfrak{g}$-valued connection. It defines the gauge-covariant derivative $D$, which acts on $\mathfrak{g}$-valued objects as

$$D_\mu(\cdot) = \nabla_\mu(\cdot) + [A_\mu, \cdot]\,. \tag{2.1}$$

The corresponding field strength is

$$F_{\mu\nu} = \partial_\mu A_\nu - \partial_\nu A_\mu + [A_\mu, A_\nu]\,. \tag{2.2}$$

Note this is not quite the gauge-covariant exterior derivative of the gauge field. Under a finite gauge transformation parametrized by $G$-valued $\Lambda$ we have

$$A_\mu \to \Lambda^{-1}(A_\mu + \nabla_\mu)\Lambda\,, \qquad F_{\mu\nu} \to \Lambda^{-1}F_{\mu\nu}\Lambda\,. \tag{2.3}$$

Likewise, under an infinitesimal gauge transformation parametrized by $\mathfrak{g}$-valued $\lambda$ we have

$$\delta A_\mu = D_\mu\lambda\,, \qquad \delta F_{\mu\nu} = [F_{\mu\nu}, \lambda]\,. \tag{2.4}$$

The action is

$$S = \int_M L = \frac{-1}{4g_{\mathrm{YM}}^2} \int_M \mathrm{Tr}[F_{\mu\nu}F^{\mu\nu}]\,, \tag{2.5}$$

where we have suppressed the integration measure $d^D x \sqrt{-g}$. The coupling $g_{\mathrm{YM}}$ plays no role in most of our discussion, so for convenience we set it to unity. The Lagrangian variation can then be written as

$$\delta L = \mathrm{Tr}[\delta A_\nu D_\mu F^{\mu\nu}] - \nabla_\mu \mathrm{Tr}[F^{\mu\nu}\delta A_\nu]\,. \tag{2.6}$$

The first term gives the equation of motion,

$$D^\mu F_{\mu\nu} = 0\,. \tag{2.7}$$

From the total derivative term we read off the (pre-)symplectic potential density

$$\theta_\nu \equiv -\mathrm{Tr}[\delta A^\mu F_{\mu\nu}]\,. \tag{2.8}$$

Viewing $\delta$ as the exterior derivative on (pre-)phase space and writing $\wedge$ for the wedge product on phase space, we define the (pre-)symplectic density as

$$\omega_\nu \equiv \delta\theta_\nu = \text{Tr}[\delta A^\mu \wedge \delta F_{\mu\nu}] \,. \tag{2.9}$$

It is a one-form on spacetime and a two-form on phase space. We obtain the (pre-)symplectic form by integrating $\omega$'s flux through a Cauchy slice,

$$\Omega \equiv \int_\Sigma *\omega \,. \tag{2.10}$$

Since $d*\omega = 0$ on shell, this is independent of the choice of $\Sigma$ as long as $\partial\Sigma$ is fixed.[3] We will work on shell from now on.

Splitting the coordinates as $x^\mu = (t, x^i)$, with coordinates $x^i$ on $\Sigma$ and $g_{ti}|_\Sigma = 0$, we define the "electric" field on $\Sigma$ as

$$E_i \equiv F_{i\mu} t^\mu \tag{2.11}$$

where $t^\mu$ is the future-directed unit normal vector to $\Sigma$. The equation of motion implies the Gauss constraint

$$D_i E^i = 0 \,. \tag{2.12}$$

In terms of $E_i$ we have

$$\Omega = \int_\Sigma \text{Tr}\big[\delta A^i \wedge \delta E_i\big] \,. \tag{2.13}$$

Phase space is defined as the quotient of solution space by the degenerate directions of $\Omega$,[4] all of which will arise from gauge transformations. First note that any gauge parameter $\lambda$ restricting to zero on $\Sigma$ will constitute a degeneracy of $\Omega$. We can use these to set $A_t = 0$ everywhere on $M$. Once this is done, solutions are uniquely specified by data $A_i$, $E_i$ on $\Sigma$.

There are still time-independent residual gauge transformations, parametrized by the gauge parameter's value on $\Sigma$. Consider plugging one of these into $\Omega$. We write $I_{\hat\lambda}$ for the phase space interior product with the infinitesimal gauge transformation parametrized by $\lambda$, so for example $I_{\hat\lambda}\delta A_i = D_i\lambda$ and $I_{\hat\lambda}\delta E_i = [E_i, \lambda]$. One finds after some algebra that

$$I_{\hat\lambda}\Omega = \int_{\partial\Sigma} \text{Tr}\big[\lambda \delta E_i n^i\big] \,, \tag{2.14}$$

where $n^i$ is the outward unit normal to $\partial\Sigma$ in $\Sigma$. We see that any $\lambda$ without support on $\partial\Sigma$ gives zero and therefore constitutes a degenerate direction of $\Omega$, which must be quotiented out. We call such gauge transformations "small". In contrast, when $\lambda$ is supported on $\partial\Sigma$ it is symplectically nontrivial and we call it "large". In general the Noether (surface) charge of a (gauge) symmetry is extracted as $I_{\hat\lambda}\Omega = \delta Q[\lambda]$. The charge associated with a large gauge transformation is then

$$Q[\lambda] \equiv \int_{\partial\Sigma} \text{Tr}\big[\lambda E_i n^i\big] \,. \tag{2.15}$$

---

[3]Recall we are currently working on a causal diamond.

[4]See however [17] for another perspective.

To recapitulate, phase space is obtained by taking the space of $A_i$, $E_i$ on $\Sigma$ with $D_i E^i = 0$ and quotienting by small gauge transformations. This leaves an infinite set of physical large gauge symmetries and corresponding charges. Our primary goal in this paper is to understand to what extent the associated degrees of freedom, i.e. the edge modes, can be isolated and separated from the remaining bulk degrees of freedom.

## 3 Bulk-Edge Split

In this section we separate the degrees of freedom of Yang-Mills theory on a causal diamond $M = \mathcal{D}(\Sigma)$ into bulk and edge parts, where the bulk modes are parametrized by functions on $\Sigma$ and the edge modes are parametrized by functions on $\partial\Sigma$. In this context the boundary $\partial\Sigma$ can also be referred to as the corner, surface, or edge. In section 3.1 we demonstrate how to split the symplectic form, and in section 3.2 we invert the symplectic form to obtain the Poisson brackets among the elementary fields.

### 3.1 Symplectic Split

We now introduce a decomposition of the phase space variables $A_i$, $E_i$ that leads to a split of the symplectic form and a consequent factorization of phase space into bulk and edge parts. This decomposition is motivated by the successful decomposition made in [7] for Maxwell, and carefully improved to take into account the non-abelian nature of the theory at hand. We write

$$A_i = U^{-1}(\tilde{A}_i + \nabla_i)U \,, \qquad E_i = U^{-1}(\tilde{E}_i + S\nabla_i\beta)U \,, \qquad (3.1)$$

where the non-dynamical function $S$ is positive and real-valued, the field $U$ is $G$-valued, and the other fields are $\mathfrak{g}$-valued. We will impose certain conditions to make this decomposition unique. Then a choice of $A_i$, $E_i$ will be equivalent to a choice of $\tilde{A}_i$, $\tilde{E}_i$, $U$ on $\Sigma$ and a choice of $n^i\nabla_i\beta|_{\partial\Sigma}$ on the boundary. By construction $\tilde{A}_i$, $\tilde{E}_i$, and $\beta$ will be gauge-invariant, and under a gauge transformation by $\Lambda$ we will simply have $U \to U\Lambda$.

To start, we require the bulk fields to have vanishing boundary normals,

$$\tilde{A}_i n^i|_{\partial\Sigma} = \tilde{E}_i n^i|_{\partial\Sigma} = 0 \,. \qquad (3.2)$$

We also require the terms in $E_i$ to separately satisfy the Gauss constraint, which implies

$$\tilde{D}_i \tilde{E}^i = \tilde{D}_i(S\nabla^i\beta) = 0 \,, \qquad (3.3)$$

where $\tilde{D}_i$ is the gauge-covariant derivative with respect to $\tilde{A}_i$, as opposed to $A_i$. As long as $\tilde{A}_i$ is within the Gribov horizon [58] there is a unique solution for $\beta$,[5] with $Sn^i\nabla_i\beta|_{\partial\Sigma} = Un^i E_i U^{-1}|_{\partial\Sigma}$ as Neumann data. This in turn determines $\tilde{E}_i$, and thus the decomposition is unique. We need another condition to make $A_i$'s decomposition unique, but we leave it unspecified for now. A natural choice will arise momentarily.

---

[5]$\beta$ is unique up to a constant shift, which drops out of $\nabla_i\beta$ and is therefore unimportant. We fix this ambiguity by requiring $\int_{\partial\Sigma} \beta = 0$.

Using our decomposition and the identity

$$\delta A_i = U^{-1}\delta\tilde{A}_i U + D_i(U^{-1}\delta U)\,, \tag{3.4}$$

we can manipulate the symplectic form and obtain

$$
\begin{aligned}
\Omega &= -\delta\int_\Sigma \mathrm{Tr}\big[\delta A_i E^i\big]\\
&= -\delta\int_\Sigma \mathrm{Tr}\Big[\big(U^{-1}\delta\tilde{A}_i U + D_i(U^{-1}\delta U)\big)E^i\Big]\\
&= -\delta\int_\Sigma \mathrm{Tr}\Big[U^{-1}\delta\tilde{A}_i U E^i\Big] - \delta\int_{\partial\Sigma}\mathrm{Tr}\big[U^{-1}\delta U E_i n^i\big]\\
&= -\delta\int_\Sigma \mathrm{Tr}\Big[\delta\tilde{A}_i(\tilde{E}^i + S\nabla^i\beta)\Big] - \delta\int_{\partial\Sigma}\mathrm{Tr}\big[U^{-1}\delta U E_i n^i\big]\\
&= \int_\Sigma \mathrm{Tr}\Big[\delta\tilde{A}_i\wedge\delta\tilde{E}^i + \delta\tilde{A}_i\wedge S\nabla^i\delta\beta\Big] - \delta\int_{\partial\Sigma}\mathrm{Tr}\big[U^{-1}\delta U E_i n^i\big]\\
&= \int_\Sigma \mathrm{Tr}\Big[\delta\tilde{A}_i\wedge\delta\tilde{E}^i - \delta\big(\nabla_i(S\tilde{A}^i)\big)\wedge\delta\beta\Big] - \delta\int_{\partial\Sigma}\mathrm{Tr}\big[U^{-1}\delta U E_i n^i\big]\,.
\end{aligned}
\tag{3.5}
$$

The only obstruction to a bulk-edge split is the term $\delta\big(\nabla_i(S\tilde{A}^i)\big)$. This motivates imposing the condition

$$\nabla_i(S\tilde{A}^i) = 0\,, \tag{3.6}$$

which at a mathematical level is essentially Lorenz or Landau gauge. The question of whether or not this can be uniquely achieved is difficult [58], but beyond the scope of this paper. There is at least no issue in perturbative contexts, where Landau gauge is standard. We will proceed under the assumption that (3.6) can be imposed. Given (3.1), it is equivalent to the following condition on $U$,

$$\nabla_i(S\nabla^i U U^{-1}) = \nabla_i(SU A^i U^{-1})\,, \tag{3.7}$$

along with the Neumann data $n^i A_i|_{\partial\Sigma} = U^{-1} n^i \nabla_i U|_{\partial\Sigma}$. These determine $U$'s bulk values in terms of its boundary values, up to an ambiguity under left multiplication by any constant, $U \to \Lambda_0 U$. This is the non-abelian version of a zero mode ambiguity. This ambiguity manifests in (3.6) as $\tilde{A}_i \to \Lambda_0 \tilde{A}_i \Lambda_0^{-1}$. We assume a smooth choice of representative is made for $\tilde{A}_i$. Then $\tilde{A}_i$ and $U$ are uniquely determined by $A_i$.[6] In turn $\tilde{E}_i$ and $\beta$ are determined as well. We return to these questions about determining $\tilde{A}_i$, $U$, $\tilde{E}_i$, $\beta$ in terms of $A_i$ and $E_i$ in a perturbative context in section 5.1.

The decomposition for $A_i$ in (3.1) is now unique, and the symplectic form reduces to

$$\Omega = \int_\Sigma \mathrm{Tr}\Big[\delta\tilde{A}^i\wedge\delta\tilde{E}_i\Big] - \delta\int_{\partial\Sigma}\mathrm{Tr}\big[U^{-1}\delta U E_i n^i\big]\,. \tag{3.8}$$

Recalling that $n^i E_i|_{\partial\Sigma} = U^{-1} S n^i \nabla_i \beta U$, we see that the bulk integral involves only the bulk fields $\tilde{A}_i$, $\tilde{E}_i$ and the boundary integral involves only the edge degrees of freedom $U|_{\partial\Sigma}$ and

---

[6]For completeness we note a further subtlety, analogous to the coordinate singularity at the origin in polar coordinates. When $\tilde{A}_i = 0$ the transformation $U \to \Lambda_0 U$ does not affect $A_i$ at all. It represents a symplectic degeneracy, which is quotiented out.

$n^i \nabla_i \beta|_{\partial\Sigma}$. These edge modes are independent functions on the edge $\partial\Sigma$. The bulk extension of $\beta$ is determined by (3.3). The bulk extension of $U$ satisfies (3.7), but if we view $U$ as the independent variable then *any* bulk extension is admissible by an appropriate choice of $A_i$. However, since the symplectic form only involves $U$'s boundary values, all extensions of a given $U|_{\partial\Sigma}$ are equivalent in the symplectic reduction. This is just the familiar fact, discussed near (2.14), that small gauge transformations are symplectically trivial.

To write $\Omega$ in its final form we define

$$E_\perp \equiv n^i E_i|_{\partial\Sigma} \,, \tag{3.9}$$

and we note

$$\delta(U^{-1}\delta U) = -U^{-1}\delta U \wedge U^{-1}\delta U \,. \tag{3.10}$$

We then have

$$\Omega = \int_\Sigma \text{Tr}\Big[\delta\tilde{A}^i \wedge \delta\tilde{E}_i\Big] + \int_{\partial\Sigma} \text{Tr}\Big[U^{-1}\delta U \wedge \delta E_\perp + U^{-1}\delta U \wedge U^{-1}\delta U E_\perp\Big] \,. \tag{3.11}$$

This split of the symplectic form into bulk and edge parts is our first main result, enabled by our decomposition (3.1). Consequently the total phase space, $\Gamma$, factorizes as a direct product, $\Gamma = \Gamma_{\text{bulk}} \times \Gamma_{\text{edge}}$. This property is nontrivial, particularly for non-abelian gauge theories, where it was generally only known that $\Gamma_{\text{edge}} \subset \Gamma$. In our approach, we have successfully constructed two independent symplectic structures: one for the bulk and another for the edge. As we will show, unlike the scenario in [7], the Hamiltonian dynamics introduce a temporal intertwining of these symplectic pairs, a phenomenon dictated entirely by the non-abelianity of the gauge group.

## 3.2 Poisson Brackets

We now derive the Poisson brackets satisfied by our fields by explicitly inverting the symplectic form. The object $\delta U$ is somewhat unnatural, outputting a tangent vector at the base point $U$. In contrast, the object $U^{-1}\delta U$ outputs a tangent vector at the identity, which is canonically identified as an element of the algebra. Thus we can define the phase space one-form

$$u^a \equiv \text{Tr}\big[T^a U^{-1}\delta U\big] \,, \tag{3.12}$$

where $T^a$ is a generator of $\mathfrak{g}$ satisfying $\text{Tr}\big[T^a T^b\big] = \delta^{ab}$. Using this we define a convenient frame of one-forms on phase space,

$$e^M \equiv \Big(\delta\tilde{A}_i^a(x), \delta\tilde{E}^{i,a}(x), u^a(y), \delta E_\perp^a(y)\Big) \,. \tag{3.13}$$

In this expression and what follows, we are using $x$ for bulk points and $y$ for boundary points. Contraction of the multi-index $M$ involves summing over discrete labels and integrating over $x$ or $y$. The symplectic form in these variables reads

$$\begin{aligned}
\Omega &= \frac{1}{2}\Omega_{MN} e^M \wedge e^N \\
&= \int_\Sigma \delta\tilde{A}_i^a(x) \wedge \delta\tilde{E}^{i,a}(x) + \int_{\partial\Sigma}\left(u^a(y) \wedge \delta E_\perp^a(y) + \frac{1}{2}f^{abc}u^a(y) \wedge u^b(y) E_\perp^c(y)\right),
\end{aligned} \tag{3.14}$$

where the structure constants are defined by $[T^a, T^b] = f^{abc} T^c$. The matrix representation in terms of our frame is

$$\Omega_{MN} = \begin{pmatrix} 0 & \delta^{ab}\delta^j_i\delta(x-x') & 0 & 0 \\ -\delta^{ab}\delta^i_j\delta(x-x') & 0 & 0 & 0 \\ 0 & 0 & f^{abc}E^c_\perp(y)\delta(y-y') & \delta^{ab}\delta(y-y') \\ 0 & 0 & -\delta^{ab}\delta(y-y') & 0 \end{pmatrix}, \qquad (3.15)$$

where $\delta(x-x')$ denotes the covariant Dirac delta function. The inverse matrix is

$$\Pi^{MN} = \begin{pmatrix} 0 & -\delta^{ab}\delta^j_i\delta(x-x') & 0 & 0 \\ \delta^{ab}\delta^i_j\delta(x-x') & 0 & 0 & 0 \\ 0 & 0 & 0 & -\delta^{ab}\delta(y-y') \\ 0 & 0 & \delta^{ab}\delta(y-y') & f^{abc}E^c_\perp(y)\delta(y-y') \end{pmatrix}. \qquad (3.16)$$

One confirms this by explicitly computing $\Pi^{MN}\Omega_{NP} = \delta^M{}_P$.

The Poisson bracket on functionals $F$, $G$ on phase space is defined by

$$\{F, G\} = \Pi^{MN} e_M[F] e_N[G], \qquad (3.17)$$

where $e_M$ is the frame of vectors dual to $e^M$. Explicitly we have

$$e_M = \left( \frac{\delta}{\delta\tilde{A}^a_i(x)}, \frac{\delta}{\delta\tilde{E}^{i,a}(x)}, U(y)T^a, \frac{\delta}{\delta E^a_\perp(y)} \right), \qquad (3.18)$$

where $T^a$ is viewed as a tangent vector at the identity in the copy of $G$ at $y$. We evaluate $e_M[F]$ using the natural action of vectors on functions. For example $\frac{\delta}{\delta\tilde{A}^a_i(x)}[\tilde{A}^b_j(x')] = \delta^{ab}\delta^i_j\delta(x-x')$ and $(U(y)T^a)[U(y')] = U(y)T^a\delta(y-y')$. The non-vanishing Poisson brackets among the elementary fields are

$$\begin{aligned} \{\tilde{E}^{i,a}(x), \tilde{A}^b_j(x')\} &= \delta^{ab}\,\delta^i_j\,\delta(x-x'), \\ \{E^a_\perp(y), U(y')\} &= U(y)T^a\,\delta(y-y'), \\ \{E^a_\perp(y), E^b_\perp(y')\} &= f^{abc}E^c_\perp(y)\,\delta(y-y'). \end{aligned} \qquad (3.19)$$

As anticipated, the bulk and edge degrees of freedom do not talk to each other. The Poisson brackets of the edge modes among themselves have appeared in (2.39-41) of [15], but the vanishing of the brackets between bulk and edge modes is new to the literature, and is a nontrivial consequence of our particular choice of decomposition in (3.1).

Note that the Poisson bracket with the charge

$$Q[\lambda] = \int_{\partial\Sigma} \text{Tr}[\lambda(y)E_\perp(y)] \qquad (3.20)$$

generates a gauge transformation by $\lambda$, as it should. These charges represent the large gauge algebra as

$$\{Q[\lambda_1], Q[\lambda_2]\} = Q\big[[\lambda_1, \lambda_2]\big]. \qquad (3.21)$$

In conclusion, we have demonstrated a consistent method for decoupling the bulk symplectic structure from its edge counterpart, all while preserving a non-vanishing Noether

charge in a fully gauge-covariant framework. This approach not only resolves a crucial structural challenge but also lays the groundwork for significant advancements in the quantization of the theory. In particular, it holds the potential to address longstanding questions about the factorization of the Hilbert space, and may instruct us on other non-abelian theories such as gravity. These developments form a cornerstone of our ongoing research agenda, opening new avenues for exploration in the interplay between gauge theories, symmetries, and the covariant phase space.

## 4 Timelike Boundary

To understand how the dynamics impacts the kinematic split of the symplectic form, we consider in this section a timelike boundary and study the effects of our field decomposition on the Hamiltonian evolution. We begin in section 4.1 by reviewing and adapting to our non-abelian setting the dynamical edge mode boundary condition of [7], then in section 4.2 we discuss how the Hamiltonian gives rise to a term mixing the bulk and edge physics.

### 4.1 Dynamical Edge Mode Boundary Condition

We now consider placing Yang-Mills theory on a Lorentzian manifold $M$ whose boundary $\partial M$ is timelike, rather than null. For simplicity we assume that $M$ is static with metric

$$ds^2 = g_{tt}dt^2 + g_{ij}dx^i dx^j \,, \tag{4.1}$$

and that $\partial_t$ lies tangent to $\partial M$. The theory requires a boundary condition to be well-defined on this manifold. Consider the variation of the action around a solution to the equation of motion (2.7),

$$\text{on-shell:} \qquad \delta S = \int_{\partial M} \text{Tr}[\delta A^\mu F_{\mu\nu} n^\nu] \,. \tag{4.2}$$

Here $n^\mu$ is the outward unit normal vector to $\partial M$. If $\delta S$ does not vanish then we do not have a true saddle, and the theory is said to be variationally ill-defined.[7]

This can be remedied by imposing a boundary condition. An obvious option is to require the pullback of the gauge field $A$ to vanish on the boundary. That is, writing $i_{\partial M}$ for the pullback to $\partial M$,

$$\text{PEC:} \qquad i_{\partial M} A = 0 \,. \tag{4.3}$$

This is a minor generalization of the perfectly electrically conducting (PEC) or "relative" boundary condition in electromagnetism. Another obvious choice is to require the normal components of the field strength to vanish, or equivalently that the pullback of the dual field strength vanishes,

$$\text{PMC:} \qquad i_{\partial M} *F = 0 \,. \tag{4.4}$$

This generalizes the perfectly magnetically conducting (PMC) or "absolute" boundary condition in electromagnetism.

---

[7]This is closely related to the symplectic form being independent of deformations of the Cauchy surface's boundary $\partial\Sigma$.

However, neither PEC nor PMC gives rise to edge modes: The PEC boundary condition disallows large gauge transformations, while the PMC boundary condition sets $E_\perp = 0$. The crucial observation in [7], which also applies here, is that one can define a new "dynamical edge mode" (DEM) boundary condition allowing both $E_\perp$ and large gauge transformations,

$$\text{DEM:} \qquad A_t|_{\partial M} = 0 = n^\mu F_{\mu i}|_{\partial M} \,. \tag{4.5}$$

This is essentially PEC for the time component and PMC for the spatial components. Through an analysis nearly identical to that in section 3.1 one can show that the resulting phase space splits into bulk and edge parts. Furthermore, in this case the bulk phase space is precisely what one would get from the PMC boundary condition, so we can write

$$\Gamma_{\text{DEM}} = \Gamma_{\text{PMC}} \times \Gamma_{\text{edge}} \,, \tag{4.6}$$

where the edge phase space is symplectomorphic to the causal diamond one.

Different boundary conditions are appropriate in different contexts, but the DEM boundary condition appears to be natural whenever edge modes are expected to play a role. This includes the calculation of entanglement entropy, the characterization of subregions, and the calculation of partition functions in the presence of horizons. Furthermore it was argued in [7] that in Maxwell theory the DEM boundary condition is shrinkable,[8] and it is highly plausible that this important property also holds in Yang-Mills.

## 4.2   Failure of the Hamiltonian Split

Unlike in the abelian case, our decomposition (3.1) will not lead to a clean bulk-edge split of the Hamiltonian. The unit time vector is $\sqrt{-g^{tt}}\partial_t$, so the Hamiltonian generating $S^{-1}\sqrt{-g^{tt}}\partial_t$ on a constant time slice $\Sigma$ is

$$H = \int_\Sigma \frac{1}{S} \operatorname{Tr}\left[ \frac{1}{2} E_i E^i + \frac{1}{4} F_{ij} F^{ij} \right]. \tag{4.7}$$

Recalling the decomposition of the electric field in (3.1), and noting

$$U F_{ij} U^{-1} = \partial_i \tilde{A}_j - \partial_j \tilde{A}_i + [\tilde{A}_i, \tilde{A}_j] \equiv \tilde{F}_{ij} \,, \tag{4.8}$$

we see that the Hamiltonian is completely independent of $U$. This makes sense since it is gauge-invariant. Using the decomposition (3.1) we can proceed further and obtain

$$
\begin{aligned}
H &= \int_\Sigma \operatorname{Tr}\left[ \frac{1}{2S} \tilde{E}_i \tilde{E}^i + \tilde{E}^i \nabla_i \beta + \frac{1}{2} S (\nabla_i \beta)(\nabla^i \beta) + \frac{1}{4S} \tilde{F}_{ij} \tilde{F}^{ij} \right] \\
&= \int_\Sigma \operatorname{Tr}\left[ \frac{1}{2S} \tilde{E}_i \tilde{E}^i + \frac{1}{4S} \tilde{F}_{ij} \tilde{F}^{ij} - \beta \nabla_i \tilde{E}^i - \frac{1}{2} \beta \nabla_i (S \nabla^i \beta) \right] + \frac{1}{2} \int_{\partial \Sigma} \operatorname{Tr}\left[ \beta S n^i \nabla_i \beta \right].
\end{aligned}
\tag{4.9}
$$

---

[8]A boundary condition is said to be shrinkable if when applied to an infinitesimally small Euclidean hole the result is as if there were no hole at all. See [7] for more details.

Defining bulk, cross, and edge terms as

$$H_{\text{bulk}} \equiv \int_{\Sigma} \frac{1}{S} \text{Tr} \left[ \frac{1}{2} \tilde{E}_i \tilde{E}^i + \frac{1}{4} \tilde{F}_{ij} \tilde{F}^{ij} \right],$$

$$H_{\text{cross}} \equiv - \int_{\Sigma} \text{Tr} \left[ \beta \nabla_i \tilde{E}^i + \frac{1}{2} \beta \nabla_i (S \nabla^i \beta) \right], \tag{4.10}$$

$$H_{\text{edge}} \equiv \frac{1}{2} \int_{\partial\Sigma} \text{Tr} \left[ \beta S n^i \nabla_i \beta \right] = \frac{1}{2} \int_{\partial\Sigma} \text{Tr} \left[ \beta (U E_\perp U^{-1}) \right],$$

we have

$$H = H_{\text{bulk}} + H_{\text{cross}} + H_{\text{edge}}. \tag{4.11}$$

The cross term can be further simplified using the Gauss constraint,

$$H_{\text{cross}} = \int_{\Sigma} \text{Tr} \left[ \beta [\tilde{A}_i, \tilde{E}^i] + \frac{1}{2} \beta [\tilde{A}_i, S \nabla^i \beta] \right]. \tag{4.12}$$

Since the number of derivatives is now minimal, $H_{\text{cross}}$ cannot be simplified further. Thus we conclude that it is non-vanishing, demonstrating that the bulk and edge degrees of freedom in Yang-Mills theory are dynamically coupled. Physically, this implies that while the kinematic phase space can be decomposed into bulk and edge components on each Cauchy slice, the dynamics inherently intertwines these two sectors. As a result, the decomposition of bulk and edge fields on an evolved slice will incorporate a mixing of the fields from the previous slice, driven by the presence of the cross term in the Hamiltonian. Notably, since this term is entirely expressed through commutators, its origin can be traced to the non-abelian structure of the gauge group. In contrast, this coupling is absent in Maxwell theory [7].

The edge Hamiltonian in terms of $\beta$ is $H_{\text{edge}} = \frac{1}{2} \int_{\partial\Sigma} \text{Tr} \left[ \beta S n^i \nabla_i \beta \right]$. From the condition (3.3) one can in principle deduce $n^i \nabla_i \beta|_{\partial\Sigma}$ from $\beta|_{\partial\Sigma}$ and vice versa. To formalize this let us define an operator $K$ on $\partial\Sigma$ by

$$K\beta|_{\partial\Sigma} \equiv S n^i \nabla_i \beta|_{\partial\Sigma}. \tag{4.13}$$

The edge Hamiltonian can then be written

$$H_{\text{edge}} = \frac{1}{2} \int_{\partial\Sigma} \text{Tr}[\beta K \beta] = \frac{1}{2} \int_{\partial\Sigma} \text{Tr} \left[ (K\beta) \frac{1}{K} (K\beta) \right], \tag{4.14}$$

where $\frac{1}{K}$ is the inverse of $K$. However, one must be careful with these simple-looking expressions. Since $K$ depends implicitly on $\tilde{A}_i$, the edge Hamiltonian is not truly independent of the bulk degrees of freedom. Along with the cross term, this obstructs a bulk-edge split of the dynamics.

## 5   Special Cases

In this section we consider some special cases of our construction, and study whether simplifications occur. We start by analyzing the weak field limit, where we observe that the dynamics decouples into bulk and edge parts, since the cross term is subleading and the edge Hamiltonian's leading term is free of bulk influence. We then probe the horizon limit, where we demonstrate that the dynamical bulk-edge mixing persists.

## 5.1 Weak Field Expansion

We now consider the weak field expansion. Since the action in (2.5) is related to the action with a canonically normalized kinetic term by the field redefinition $A_\mu \to g_{\mathrm{YM}} A_\mu$, the weak field expansion is equivalent to the small coupling expansion. For clarity we will restore the coupling in this subsection. We expand our fields order by order as

$$
\begin{aligned}
\tilde{A}_i &= g_{\mathrm{YM}} \tilde{A}_i^{(1)} + g_{\mathrm{YM}}^2 \tilde{A}_i^{(2)} + \dots, \\
\tilde{E}_i &= g_{\mathrm{YM}} \tilde{E}_i^{(1)} + g_{\mathrm{YM}}^2 \tilde{E}_i^{(2)} + \dots, \\
\beta &= g_{\mathrm{YM}} \beta^{(1)} + g_{\mathrm{YM}}^2 \beta^{(2)} + \dots.
\end{aligned}
\tag{5.1}
$$

The field $U$ is slightly special, since small $A_i$ does not necessarily imply that $U$ itself is near the identity. Rather $U^{-1} \nabla_i U$ must be small. We can then use a constant $U_0 \in G$ and a $\mathfrak{g}$-valued field $\alpha$ to parametrize $U = U_0 e^\alpha$, with the expansion

$$
\alpha = g_{\mathrm{YM}} \alpha^{(1)} + g_{\mathrm{YM}}^2 \alpha^{(2)} + \dots.
\tag{5.2}
$$

The phase space splits exactly, so it will continue to split at every order in the weak field expansion. The situation for the Hamiltonian is more interesting. The bulk and edge parts in (4.10) contain terms quadratic in the fields, while the cross term is cubic in the fields and so it is relatively suppressed,

$$
H = H_{\mathrm{bulk}} + H_{\mathrm{cross}} + H_{\mathrm{edge}} = g_{\mathrm{YM}}^2 (H_{\mathrm{bulk}}^{(2)} + H_{\mathrm{edge}}^{(2)}) + \mathcal{O}(g_{\mathrm{YM}}^3).
\tag{5.3}
$$

Furthermore, as we will demonstrate shortly, $H_{\mathrm{edge}}^{(2)}$ depends only on the boundary data. Therefore there is an emergent split in the weak field limit, reducing to the abelian case. This limit is often physically relevant. For example in the perturbative evaluation of a Euclidean partition function the one-loop correction involves only the quadratic terms in the action, or equivalently in the Hamiltonian. We therefore reach an important feature of Yang-Mills theory in the weak field expansion: The bulk and edge fields are symplectically and dynamically completely disentangled.

We now revisit the conditions defining $U$ and $\beta$. Implicit in our decomposition (3.1) is the idea that, given $A_i$ and $E_i$, one should be able to deduce $\tilde{A}_i$, $U$, $\tilde{E}_i$, and $\beta$. In principle this can be accomplished by first using (3.7) to solve for $U$ in terms of $A_i$,[9] then stripping off $U$ from $A_i$ to get $\tilde{A}_i$, then using (3.3) to solve for $\beta$ in terms of $\tilde{A}_i$ and $E_i$, and finally stripping off $U$ and $\beta$ from $E_i$ to get $\tilde{E}_i$. We are now in a position to comment on the existence of solutions at each step in this procedure. The condition on $U$ has the expansion

$$
\begin{aligned}
0 &= \nabla^i (S \nabla_i e^\alpha e^{-\alpha}) - \nabla^i (S e^\alpha A_i e^{-\alpha}) \\
&= g_{\mathrm{YM}} \nabla^i \left( S \nabla_i \alpha^{(1)} - S A_i^{(1)} \right) \\
&\quad + g_{\mathrm{YM}}^2 \nabla^i \left( S \nabla_i \alpha^{(2)} + \frac{1}{2} S [\alpha^{(1)}, \nabla_i \alpha^{(1)}] - S A_i^{(2)} - S [\alpha^{(1)}, A_i^{(1)}] \right) + \mathcal{O}(g_{\mathrm{YM}}^3).
\end{aligned}
\tag{5.4}
$$

---

[9]Recall this leaves an ambiguity by constant left multiplication $U \to \Lambda_0 U$, which we choose to fix with some arbitrary smooth choice of representatives.

Note that $U_0$ has dropped out completely, due to the above-mentioned ambiguity under left multiplication by a constant. The $\mathcal{O}(g_{\mathrm{YM}})$ term is the same sourced elliptic PDE found in [7] for the Maxwell case. It uniquely determines $\alpha^{(1)}$ up to a constant shift, with $n^i A_i^{(1)}|_{\partial\Sigma}$ as Neumann data. The only unknown part of the $\mathcal{O}(g_{\mathrm{YM}}^2)$ term is then $\alpha^{(2)}$, and we find the same type of elliptic PDE for it, with Neumann data

$$n^i \nabla_i \alpha^{(2)}|_{\partial\Sigma} = n^i \left( A_i^{(2)} + \frac{1}{2}[\alpha^{(1)}, \nabla_i \alpha^{(1)}] \right)\Big|_{\partial\Sigma}. \tag{5.5}$$

This pattern of PDEs determining $\alpha^{(n)}$ in terms of lower order data continues to all perturbative orders, fully determining $\alpha$'s expansion up to zero modes. These zero modes and $U_0$ are then fully determined by the choice of representative. Therefore $U$ is perturbatively well-defined.

With $U$ in hand, $\tilde{A}_i$ is determined as

$$\tilde{A}_i = U A_i U^{-1} - \nabla_i U U^{-1}. \tag{5.6}$$

The condition (3.3) on $\beta$ can be expanded as

$$
\begin{aligned}
0 &= \tilde{D}_i(S\nabla^i\beta) \\
&= \nabla_i(S\nabla^i\beta) + [\tilde{A}_i, S\nabla^i\beta] \\
&= g_{\mathrm{YM}}\nabla_i(S\nabla^i\beta^{(1)}) + g_{\mathrm{YM}}^2\left( \nabla_i(S\nabla^i\beta^{(2)}) + [\tilde{A}_i^{(1)}, S\nabla^i\beta^{(1)}] \right) + \mathcal{O}(g_{\mathrm{YM}}^3),
\end{aligned}
\tag{5.7}
$$

giving a series of PDEs for $\beta^{(n)}$. The Neumann data at each order follow from

$$U n^i E_i U^{-1}\big|_{\partial\Sigma} = S n^i \nabla_i \beta|_{\partial\Sigma}. \tag{5.8}$$

The zero modes of the $\beta^{(n)}$ are undetermined, but they are unimportant since they drop out of $\nabla_i\beta$, which is what actually shows up in the field decomposition. We fix this ambiguity by choosing $\int_{\partial\Sigma} \beta^{(n)} = 0$. Now $\beta$ is perturbatively well-defined. Note in particular that the leading order condition on $\beta$ does not involve $\tilde{A}_i$, so there are no bulk contributions to the relationship between $\beta^{(1)}$ and its normal derivative. This shows that $H_{\mathrm{edge}}^{(2)} = \frac{1}{2}\int_{\partial\Sigma} \mathrm{Tr}\big[\beta^{(1)} S n^i \nabla_i \beta^{(1)}\big]$ depends only on boundary data, as mentioned above. It only remains to define the bulk electric field,

$$\tilde{E}_i = U E_i U^{-1} - S \nabla_i \beta. \tag{5.9}$$

This completes the process of deducing $\tilde{A}_i$, $U$, $\tilde{E}_i$, $\beta$ from $A_i$ and $E_i$.

## 5.2 Horizon Limit

We now study the horizon limit of our timelike boundary. The setup is as in section 2.5 of [7]. Specifically, consider a static manifold whose boundary is a static bifurcate horizon and let our $M$ be the subregion whose boundary is at a small spatial distance $\varepsilon$ from the horizon. Within each time slice $\Sigma$ we establish Gaussian normal coordinates in a neighborhood of the horizon, so that the full metric is

$$ds^2 = g_{tt}dt^2 + dr^2 + g_{ab}dx^a dx^b. \tag{5.10}$$

Here $r$ is the normal coordinate, with $r = 0$ on the bifurcation surface and $r = \varepsilon$ on $\partial M$. The $x^a$ are coordinates on $\partial\Sigma$. Our assumption of a static bifurcate horizon implies

$$g_{tt} = -\kappa^2 r^2 + \mathcal{O}(r^4) \,, \tag{5.11}$$

where $\kappa$ is the surface gravity. In this subsection we will take $S = \sqrt{-g^{tt}}$, corresponding to the Hamiltonian generating $\partial_t$.

We have shown that the bulk and edge degrees of freedom are dynamically coupled at finite $\varepsilon$, but we would like to know if this mixing persists in the horizon limit. We will tackle this question in the weak field approximation. The leading term of $H_{\text{cross}}$ is already $\mathcal{O}(g_{\text{YM}}^3)$,

$$H_{\text{cross}} = g_{\text{YM}}^3 \int_\Sigma \text{Tr}\left[\beta^{(1)}[\tilde{A}_i^{(1)}, \tilde{E}^{(1),i}] + \frac{1}{2}\beta^{(1)}[\tilde{A}_i^{(1)}, \sqrt{-g^{tt}}\nabla^i\beta^{(1)}]\right] + \mathcal{O}(g_{\text{YM}}^4) \,. \tag{5.12}$$

Note this only involves the $\mathcal{O}(g_{\text{YM}})$ parts of the fields, which obey the same PDEs as in the abelian case, and so their properties can be borrowed from [7]. In particular we have

$$\beta^{(1)} = \mathcal{O}\left(\frac{r^0}{\log \varepsilon^{-1}}\right), \qquad \sqrt{-g^{tt}}\nabla_r\beta^{(1)} = \mathcal{O}\left(\frac{\log r^{-1}}{\log \varepsilon^{-1}}\right), \qquad \sqrt{-g^{tt}}\nabla_a\beta^{(1)} = \mathcal{O}\left(\frac{r^{-1}}{\log \varepsilon^{-1}}\right), \tag{5.13}$$

as well as

$$\tilde{E}_r^{(1)} = \mathcal{O}(r\,\varepsilon^0) \,, \qquad \tilde{E}_a^{(1)} = \mathcal{O}(r^0\varepsilon^0) \,, \tag{5.14}$$

and

$$\tilde{A}_r^{(1)} = \mathcal{O}(r\,\varepsilon^0) \,, \qquad \tilde{A}_a^{(1)} = \mathcal{O}(r^0\varepsilon^0) \,. \tag{5.15}$$

The main observation is that the leading behavior of the Hamiltonian cross term is

$$H_{\text{cross}}^{(3)} = \mathcal{O}\left(\frac{1}{\log \varepsilon^{-1}}\right) \,, \tag{5.16}$$

which, as we show in appendix A, is of the same order as $H_{\text{edge}}^{(3)}$. This behavior is readily seen in the first term of (5.12). The second term in $H_{\text{cross}}$ also contributes with this leading behavior, which can be deduced from the part of the integral near the boundary.[10]

In conclusion we have found that $H_{\text{cross}}^{(3)}$ is of the same order as $H_{\text{edge}}^{(3)}$, and therefore is not relatively suppressed in the horizon limit. This is a clear indication that the dynamical mixing of the bulk and edge degrees of freedom persists in the horizon limit.[11]

## 6   Final Words

In this paper we discussed a suitable decomposition of the dynamical fields — the electric field and the spatial gauge field — in Yang-Mills theory on a Cauchy surface with boundary. We found an appropriate way to disentangle the bulk fields from the edge modes, such that

---

[10]Specifically one uses $\int_\varepsilon \frac{dr\, r^{-1}}{(\log \varepsilon^{-1})^2} = \mathcal{O}\left(\frac{1}{\log \varepsilon^{-1}}\right)$.

[11]In fact, there is also mixing of bulk and boundary data in $H_{\text{edge}}^{(3)}$ itself, due to the $\tilde{A}_i$ dependence in the operator $K$.

the symplectic structure splits and thus the phase space factorizes. With this decomposition, we computed the Poisson brackets and showed that the bulk and edge degrees of freedom do not talk to each other. That is, bulk fields Poisson-commute with edge fields. We then switched our attention to timelike boundaries, and discussed how to implement the DEM boundary condition proposed for Maxwell theory in [7]. Contrary to Maxwell theory, we showed that the Yang-Mills Hamiltonian couples the bulk and edge contributions, due to a cross term. We rewrote this cross contribution in terms of commutators to emphasize that it is entirely due to the non-abelianity of the gauge group. Indeed, we then provided a weak field expansion in $g_{\mathrm{YM}}$, in which we demonstrated that the cross term in the Hamiltonian vanishes at leading order. We then discussed the horizon limit of the timelike boundary. Here the cross term survives, leading to a mixed evolution that survives the limit.

There are many future avenues of investigation for which this work sets the stage. First, a careful analysis of the partition function and entanglement structure of Yang-Mills theory with the DEM boundary condition remains to be done, and is currently under investigation. The one-loop partition function uses only the quadratic part of the action, and so it essentially reduces to the abelian case. At higher loops the mixed term in the Hamiltonian poses an interesting challenge, and we may anticipate that it will lead to important modifications with respect to the abelian analysis.

Another interesting road to explore is the relationship between this work and Yang-Mills theory in Minkowski space. First of all, we wish to understand how the DEM boundary condition and the symplectic split relates to the asymptotic symmetries of Yang-Mills [59–62] and Einstein-Yang-Mills [63, 64]. Secondly, we intend to study how the field decomposition proposed here intertwines with the celestial holographic description [65–69]. Lastly, we ask if our work informs the soft split [70, 71] and gluon soft theorems [72–75].

The most important future direction to explore is gravity. While there has been much progress in recent years in the understanding of gravitational edge modes [23–25, 27–35], it would be rewarding to study the field decomposition and achieve a complete symplectic split among bulk and corner fields. To the best of our knowledge, this is still missing in the literature, and could constitute an important step forward in the corner proposal [15, 36–40]. We expect that our main results in this manuscript will extend to gravitational theories, and we intend to study exactly how this can be achieved. Indeed, the fact that the symplectic structure still splits when non-abelian effects are taken into account was an important cornerstone of this paper, and paves the way for a similar pattern in gravity. Furthermore, the fact that the gravitational Hamiltonian is already a boundary term indicates that such a symplectic split should be preserved in the gravitational dynamics. In conclusion, the journey from Maxwell theory [7] to gravity had to pass through Yang-Mills theory. This work has filled that gap, preparing the ground for future explorations.

## Acknowledgments

We thank Glenn Barnich, Laurent Freidel, Rob Leigh, and Aldo Riello for useful discussions and feedback. AB is supported by the Celestial Holography Initiative at the Perimeter Institute for Theoretical Physics and the Simons Collaboration on Celestial Holography.

Research at Perimeter Institute is supported in part by the Government of Canada through the Department of Innovation, Science and Economic Development Canada and by the Province of Ontario through the Ministry of Colleges and Universities.

# A  Weak and Horizon Limit of $H_{\text{edge}}$

In this appendix we show that in the simultaneous weak field and horizon limits the edge Hamiltonian's first subleading term in $g_{\text{YM}}$ is of order

$$H_{\text{edge}}^{(3)} = \mathcal{O}\Big(\frac{1}{\log \varepsilon^{-1}}\Big). \tag{A.1}$$

We start from (4.10),

$$H_{\text{edge}} = \frac{1}{2} \int_{\partial\Sigma} \text{Tr}\big[\beta(U E_\perp U^{-1})\big]. \tag{A.2}$$

Both $E_\perp$ and $U$ are independent of $\varepsilon$, which follows from the fact that their path integral measure can be read off from the symplectic form, and that $\Omega_{\text{edge}}$ is manifestly independent of $\varepsilon$. Considering also that $U E_\perp U^{-1}$ starts at order $\mathcal{O}(g_{\text{YM}})$, we see that the $\varepsilon$ dependence of $H_{\text{edge}}^{(3)}$ will come entirely from $\beta^{(2)}$.

Consider the horizon limit of the weak field expansion (5.7) of the condition satisfied by $\beta$. The leading term is

$$0 = \nabla_i(\sqrt{-g^{tt}}\nabla^i\beta^{(1)}). \tag{A.3}$$

It was shown in [7] that this PDE admits the separation of variables

$$\beta^{(1)}(r, x^a) = \sum_{k\neq 0} \beta_k^{(1)}(r) f_k(x^a), \tag{A.4}$$

where the $f_k(x^a)$ are orthonormal eigenmodes of the Laplacian on $\partial\Sigma$, with non-negative eigenvalues $\Delta_{\partial\Sigma} f_k(x^a) = \lambda_k f_k(x^a)$, and we omit the zero mode from the sum since it drops out of the PDE. The asymptotic solution near the horizon was shown to be

$$\beta_k^{(1)}(r) \propto 1 + \frac{1}{2}\lambda_k r^2 \log\sqrt{\lambda_k}r + \mathcal{O}(r^2), \tag{A.5}$$

from which one can deduce that the operator $K$, defined in (4.13), reduces in the $\varepsilon \to 0$ limit to

$$(K\beta)^{(1)} = \frac{\log\varepsilon^{-1}}{\kappa}\Delta_{\partial\Sigma}\beta^{(1)}. \tag{A.6}$$

Since the quantity $(U E_\perp U^{-1})^{(1)}$ is independent of $\varepsilon$, this implies that $\beta^{(1)} = \mathcal{O}(\frac{1}{\log\varepsilon^{-1}})$.

The first subleading term in $\beta$'s condition (5.7) is

$$0 = \nabla_i(\sqrt{-g^{tt}}\nabla^i\beta^{(2)}) + [\tilde{A}_i^{(1)}, \sqrt{-g^{tt}}\nabla^i\beta^{(1)}]. \tag{A.7}$$

This is a PDE for $\beta^{(2)}$, similar to the previous PDE for $\beta^{(1)}$ except that now it is sourced by the commutator term. Although $\tilde{A}_r^{(1)} = \mathcal{O}(r)$ due to the condition $n^i\tilde{A}_i|_{\partial\Sigma} = 0$, the other components $\tilde{A}_a^{(1)}$ are not small. Overall the commutator term is of order $\mathcal{O}(\frac{1}{\log\varepsilon^{-1}})$, i.e. the same size as $\beta^{(1)}$, and so the inhomogeneous part of $\beta^{(2)}$ will be of this same size.

Note this means $\beta^{(2)}$ is affected by $\tilde{A}_a^{(1)}$, indicating some dynamical mixing of the bulk and edge degrees of freedom, as mentioned near (4.14).

The homogeneous part of $\beta^{(2)}$ is determined by the Neumann data

$$\sqrt{-g^{tt}}n^i\nabla_i\beta^{(2)}|_{\partial\Sigma} = (K\beta)^{(2)} = (UE_\perp U^{-1})^{(2)}\,. \tag{A.8}$$

Once again neither $E_\perp$ nor $U$ scales with $\varepsilon$, so the homogeneous part of $\beta^{(2)}$ is $\mathcal{O}(\frac{1}{\log\varepsilon^{-1}})$, just like for $\beta^{(1)}$. Then overall

$$\beta^{(2)} = \mathcal{O}(\frac{1}{\log\varepsilon^{-1}})\,, \tag{A.9}$$

and in turn we have the desired property (A.1).

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
