# Peer review of "Dynamical Edge Modes in Yang-Mills Theory"

_SciPost Physics_

## Round 1 · Referee Report · Anonymous (Referee 1) · 2025-7-12

Report

This work presents a cogent analysis of edge modes in Yang-Mills gauge theory, generalizing previous work on Maxwell theory. The authors provides a phase space analysis of Yang Mills on a manifold with boundary with a ``dynamical edge mode boundary condition". This leads to edge degrees of freedom associated to large gauge transformations. The main result is a decomposition of the Yang Mills into bulk and edge degrees of freedom, which are decouple in the symplectic form. However, unlike the case of Maxwell theory, the Hamiltonian couples the edge and bulk degrees of freedom. The authors show that this coupling of bulk and edge is absent in the weak field approximation, but persists in the near horizon limit.

Compared to previous treatments of edge modes using the covariant phase space formalism, the analysis produces more dynamical information because of the introduction of a time like boundary. In particular, an explicit expression for the Hamiltonian is given. This paves the way for computations related to the modular Hamiltonian and entanglement entropy of a subregion.

The paper is very clearly written, and provides an important step towards the understanding of the entanglement structure of non abelian gauge theories. The potential for applications to asymptotic soft modes and generalizations to gravity is tantalizing. I believe it meets the journal's criteria for originality and relevance to an exciting field of research.

Recommendation

Publish (surpasses expectations and criteria for this Journal; among top 10%)

  • validity: high
  • significance: high
  • originality: high
  • clarity: high
  • formatting: excellent
  • grammar: excellent

Author:  Luca Ciambelli  on 2025-11-11  [id 6009]

(in reply to Report 1 on 2025-07-12)

We would like to thank the referee for their positive and constructive report.

---

## Round 1 · Referee Report · Anonymous (Referee 2) · 2025-11-1

Report

The paper discusses edge modes in YM theory. There is quite some litterature on this topic, but most of it is rather hard to read. This paper contains a very clear presentation, which moreover contains several new results.
The main results are equation (3.8) and (3.19), and (4.10).
The decomposition of phase space as (3.8) and associated (3.19) are intuitive and perhaps more or less expected, nevertheless they have never been presented this explicitly (and certainly not with such a clear derivation). This is a most welcome addition to the litterature.
I found the mixing in section 5 not very surprising, because a timelike boudnary does not define a causal domain, so one would expect in general things to pass through the boundary and thus bulk and edge to be mixed.

Requested changes

I have two questions/comments 1. It would be nice to see the relation of the DEM boundary conditions in section 4.1 with the split discussed in section 3, as presented their relation did not seem very obvious to me. 2. Intuitively I have problems with saying that there is dynamical mixing when the boundary is a horizon. Time evolution does not act at the horizon (because it is a point or codim two surface), suggesting that edge modes on a horizon should have in some sense no energy. It was my naive understanding that such an interpretation existsed purely for the edge Hamiltonian. Since the authors argue that the mix Hamiltonian is of the same order as the edge Hamiltonian, I would suspect or hope that the same is true for the mixing term. How can a degree of freedom that is living on a horizon with no time coordinate, evolve in that time coordinate? A possible explanation could be the bulk extention of the beta field which could appear perhaps as a time-independent source term in the evolution of the bulk fields. It would be appreciated it the authors could share their thoughts about this.

Recommendation

Ask for minor revision

  • validity: top
  • significance: high
  • originality: ok
  • clarity: top
  • formatting: excellent
  • grammar: perfect

Author:  Luca Ciambelli  on 2025-11-11  [id 6010]

(in reply to Report 2 on 2025-11-01)

The referee raises interesting points which we are glad to be able to clarify.

  • Comment 1

The discussion around (4.5) aimed at addressing this question. The split discussed in section 3 is structurally independent of the chosen boundary conditions: different boundary conditions determine whether the edge contribution is present or happens to vanish. So boundary conditions do not inform the split construction. Instead, the split construction allows us to distinguish among different boundary conditions, as they confer different conditions on the split data.

To emphasize this, we have amended the sentence below (4.5)

It originally read

Through an analysis nearly identical to that in section 3.1 one can show that the resulting phase space splits into bulk and edge parts. Furthermore, in this case the bulk phase space is precisely what one would get from the PMC boundary condition, so we can write

We have amended it into

With this choice of boundary conditions, the analysis performed in section 3.1 leads to a bulk phase space which is precisely what one would get from the PMC boundary condition, so we can write...

  • Comment 2

This is indeed a subtle point. As we take our boundary to hug the horizon, one has H_edge -> 0. Nonetheless, we then comment on dynamical mixing at the end of that section. An important feature, which we have not mentioned in the manuscript (see below for a proposed improvement) is that, while indeed the edge piece of the Hamiltonian goes to zero in the limit, there is still a non-trivial contribution to the entropy coming from the edge part. We have not commented on that here because we have not reproduced this result for Yang-Mills. However, this was already observed for Maxwell in reference [7], and we have sufficient evidence that this feature persists in Yang-Mills too. The technical reason why the entropy still receives a contribution is not completely clear to the authors. Intuitively, it should be related to the fact that, as H_edge -> 0, one has Tr exp(-beta H_edge) -> infinity, which thus requires requires some extra regularization, with a potentially-relevant finite piece arising. \

In practice, we propose to add a footnote (footnote 12), adding the following sentence

Although the edge Hamiltonian vanishes as the boundary approaches the horizon, the edge sector still contributes to the entropy. This behavior is known in Maxwell [7] and is expected to persist in Yang-Mills, though its precise origin remains subtle. Intuitively, as H_edge -> 0, one has Tr exp(-beta H_edge) -> infinity, necessitating an additional regularization that can leave a finite, physically relevant term.

We thank the referee for the questions, which we hope we had satisfactorily addressed.

---

## Editorial Decision

resubmitted